# BEARD: BENCHMARKING THE ADVERSARIAL ROBUSTNESS FOR DATASET DISTILLATION

## ABSTRACT

Dataset Distillation (DD) is an emerging technique that compresses large-scale datasets into significantly smaller synthesized datasets while preserving high test performance and enabling the efficient training of large models. However, current research primarily focuses on enhancing evaluation accuracy under limited compression ratios, often overlooking critical security concerns such as adversarial robustness. A key challenge in evaluating this robustness lies in the complex interactions between distillation methods, model architectures, and adversarial attack strategies, which complicate standardized assessments. To address this, we introduce BEARD, an open and unified benchmark designed to systematically assess the adversarial robustness of DD methods, including DM, IDM, and BA-CON. BEARD encompasses a variety of adversarial attacks (e.g., FGSM, PGD, C&W) on distilled datasets like CIFAR-10/100 and TinyImageNet. Utilizing an adversarial game framework, it introduces three key metrics: Robustness Ratio (RR), Attack Efficiency Ratio (AE), and Comprehensive Robustness-Efficiency Index (CREI). Our analysis includes unified benchmarks, various Images Per Class (IPC) settings, and the effects of adversarial training. Results are available on the BEARD Leaderboard, along with a library providing model and dataset pools to support reproducible research. Access the code at BEARD.

## 1 INTRODUCTION

Deep Neural Networks (DNNs) (LeCun et al., 2015) have achieved significant success across various applications, primarily due to the availability of large datasets (Krizhevsky et al., 2012; Vaswani et al., 2017; Radford et al., 2021; Kirillov et al., 2023). These extensive datasets enable DNNs to learn valuable representations tailored to specific tasks. However, the acquisition of such large datasets and the training of DNNs can be prohibitively expensive.

Dataset Distillation (DD), an emerging technique that compresses large datasets into smaller sets of synthetic samples (Wang et al., 2018; Zhao et al., 2020; Cazenavette et al., 2022; Nguyen et al., 2020; Cui et al., 2023; Zhou et al., 2024a), offers a cost-effective alternative by reducing training demands and simplifying dataset acquisition. DD has a profound impact on both research and practical applications, facilitating the efficient handling and processing of vast amounts of data across various fields. Significant progress in DD has been driven by advanced algorithms, which can be categorized into Meta-Model Matching (e.g., DD (Wang et al., 2018), KIP (Nguyen et al., 2020; 2021), RFAD (Loo et al., 2022), and FRePo (Zhou et al., 2022a)), Gradient Matching (e.g., DC (Zhao et al., 2020), MTT (Cazenavette et al., 2022), TESLA (Cui et al., 2023), and FTD (Du et al., 2023)), and Distribution Matching (e.g., DM (Zhao & Bilen, 2021), CAFE (Wang et al., 2022), IDM (Zhao et al., 2023), and BACON (Zhou et al., 2024a)).

Despite these advancements, DNNs remain highly susceptible to adversarial attacks. These attacks involve perturbations that are imperceptible to the human eye but can effectively deceive classifiers when added to clean images (i.e., adversarial examples) (Szegedy et al., 2014; Goodfellow et al., 2015a; Madry et al., 2018a; Zhou et al., 2022b), as illustrated in Figure 1. Such vulnerabilities pose significant security risks in applications like face recognition (Wei et al., 2022a;b), object detection (Zhou et al., 2024b; Hu et al., 2021), and autonomous driving (Wang et al., 2021; Yuan et al., 2023), thereby undermining the reliability of DNNs.

**Research Gap.** While some studies (Wu et al., 2024; Xue et al., 2024; Ma et al., 2023; Chen et al., 2023) suggest that DD may enhance adversarial robustness, they do not fully address the complex

Figure 1: Illustration of evaluating adversarial robustness for dataset distillation: The process is divided into three stages: 1) Distillation stage, where diverse dataset distillation methods such as DC (Zhao et al., 2020), DSA (Zhao & Bilen, 2023), and DM (Zhao & Bilen, 2021) generate distilled datasets. 2) Training stage, where models are trained on these distilled datasets. 3) Evaluating stage, where adversarial attacks (e.g., FGSM (Goodfellow et al., 2015b), PGD (Madry et al., 2018b), and C&W (Carlini & Wagner, 2017)) are applied to the test set of standard datasets like CIFAR-10/100 (Krizhevsky, 2009) and TinyImageNet (Deng et al., 2009), and model performance is evaluated both with and without adversarial attacks, summarized using specific metrics.

vulnerabilities introduced by adversarial attacks. The robustness of models trained on distilled datasets has not been systematically investigated. Evaluating this robustness is uniquely challenging due to the intricate interactions between distillation methods, model architectures, and diverse attack strategies, which cannot be captured by standard attack protocols. This gap highlights the need for a rigorous framework and tailored metrics to comprehensively assess and improve the adversarial robustness of models trained on distilled data.

To address this gap, we introduce `BEARD`, an open and unified benchmark designed to systematically evaluate the adversarial robustness of existing DD methods. We conducted extensive evaluations using representative DD techniques, including DC (Zhao et al., 2020), DSA (Zhao & Bilen, 2023), DM (Zhao & Bilen, 2021), MTT (Cazenavette et al., 2022), IDM (Zhao et al., 2023), and BACON (Zhou et al., 2024a). These evaluations span a range of datasets, from large-scale collections like TinyImageNet (Deng et al., 2009) to smaller ones such as CIFAR-10/100 (Krizhevsky, 2009), encompassing diverse scenarios. We incorporated a broad spectrum of both typical and state-of-the-art attack methods for robustness evaluation, including FGSM (Goodfellow et al., 2015b), PGD (Madry et al., 2018b), C&W (Carlini & Wagner, 2017), DeepFool (Moosavi-Dezfooli et al., 2016), and AutoAttack (Croce & Hein, 2020). To thoroughly assess the adversarial robustness of DD methods, we employed the *adversarial game framework* to unify various DD tasks and attack scenarios, proposing three primary evaluation metrics: Robustness Ratio (RR), Attack Efficiency Ratio (AE), and Comprehensive Robustness-Efficiency Index (CREI). Additionally, we developed a straightforward evaluation protocol using both a Dataset Pool and a Model Pool. Our large-scale experiments involved cross-evaluating attack methods under multiple threat models, including both targeted and untargeted attacks. This analysis provides insights into adversarial robustness from the perspectives of unified benchmarks, diverse Images Per Class (IPC), and the impact of adversarial training using `BEARD`.

Our **contributions** are summarized as follows:

- We introduce `BEARD`, a unified benchmark for evaluating adversarial robustness in dataset distillation, employing an *adversarial game framework* to systematically assess DD methods under various attack scenarios.

- We propose new metrics to evaluate the adversarial robustness of distilled datasets against different attacks, accompanied by a leaderboard that ranks existing DD methods based on these metrics.

- We provide open-source code with comprehensive documentation and easy extensibility, along with a Model Pool and Dataset Pool to facilitate adversarial robustness evaluations.

- We conduct a comparative analysis of the benchmark results, offering insights and recommendations for enhancing adversarial robustness in dataset distillation.

## 2 RELATED WORK

### 2.1 DATASET DISTILLATION

Dataset Distillation (DD) synthesizes a compact set of images that preserves the key information from the original dataset. Wang et al. (2018) pioneered a bi-level optimization approach that models network parameters based on synthetic data. However, this bi-level optimization incurs additional computational costs due to its nested structure. To address these costs, Zhao et al. (2020) introduced Dataset Condensation (DC), a gradient matching method that improves performance by aligning the informative gradients from the original datasets with those from the synthetic datasets at each iteration. An enhanced version of this approach, known as DSA (Zhao & Bilen, 2023), further refines the process. Cazenavette et al. (2022) proposed mimicking the long-range training dynamics of real data by aligning learning trajectories, a method referred to as MTT. Additionally, Zhao & Bilen (2021) developed a distribution matching method called DM, which uses the Maximum Mean Discrepancy (MMD) metric. Building on DM, Zhao et al. (2023) introduced Improved Distribution Matching (IDM), a more efficient method that significantly enhances distillation performance. Zhou et al. (2024a) incorporated the Bayesian framework into DD tasks, providing robust theoretical support, further advancing distillation outcomes. Furthermore, Cui et al. (2022) introduced DC-Bench, the first benchmark for DD. Other research directions include SRe2L (Yin et al., 2024), RDED (Sun et al., 2024), multi-size dataset distillation (He et al., 2024), and lossless distillation through matching trajectories (Guo et al., 2024).

### 2.2 ADVERSARIAL DATASET DISTILLATION

Adversarial Robust Distillation (ARD) was introduced by Goldblum et al. (2020), showing that robustness can be transferred from teacher to student in knowledge distillation. Building on robust features (Ilyas et al., 2019), Wu et al. (2022) proposed creating robust datasets, ensuring classifiers trained on them exhibit inherent adversarial robustness. Subsequently, Ma et al. (2023) explored the efficiency and reliability of DD tasks with TrustDD, while Chen et al. (2023) provided a security-focused analysis of DD, highlighting associated risks. Additionally, Xue et al. (2024) investigated methods to embed adversarial robustness in distilled datasets, aiming to enhance resilience without sacrificing accuracy. Despite these contributions, the adversarial robustness of DD tasks remains underexplored. Evaluating this robustness is complicated by the intricate interactions between distillation methods, model architectures, and adversarial attacks, hindering standardized assessment. Although Wu et al. (2024) introduced a benchmark for evaluating the adversarial robustness of distilled datasets, they did not provide benchmark code or standardized metrics for resilience against various adversarial attacks.

In contrast, we introduce BEARD, an open and unified benchmark specifically designed to systematically evaluate the adversarial robustness of existing DD methods across multiple datasets. Inspired by Dai et al. (2023), we utilize an adversarial game framework to enhance this evaluation and propose three key metrics: Robustness Ratio (RR), Attack Efficiency Ratio (AE), and Comprehensive Robustness-Efficiency Index (CREI). These metrics serve as essential tools for assessing model performance under various adversarial attacks.

## 3 ADVERSARIAL ROBUSTNESS FOR DATASET DISTILLATION AGAINST MULTIPLE ATTACKS

We start with defining the notations, then analyze challenges in adversarial robustness for Dataset Distillation (DD), focusing on attack effectiveness and time efficiency. We introduce a unified adversarial game framework to tackle these issues and outline goals for improving robustness. Finally, we propose metrics within this framework to measure robustness comprehensively.

**Notations.** Consider a large dataset $\mathcal{T} = \{(x_i, y_i)\}_{i=1}^N$, where $x_i \in \mathcal{X} \subseteq \mathbb{R}^d$ denotes the input samples and $y_i \in \mathcal{Y} \subseteq \{1, \ldots, C\}$ denotes the corresponding labels. DD aims to generate a synthetic dataset $\mathcal{S} = \{(\tilde{x}_j, \tilde{y}_j)\}_{j=1}^M$ such that a model $\mathcal{M}(\cdot) : \mathcal{X} \rightarrow \mathcal{Y}$ trained on $\mathcal{S}$ performs comparably to one trained on $\mathcal{T}$. We denote the defender function, which encompasses DD methods with diverse IPC settings, as $\mathcal{D}$. The model $\mathcal{M}$ is trained on the distilled dataset generated by $\mathcal{D}$ with diverse IPC

settings $d \in \mathcal{D}$ (i.e., $\mathcal{D}$ outputs a function $m \in \mathcal{M}$). The attacker function is denoted by $\mathcal{A}$, with a perturbation budget $\epsilon \in \mathcal{P}$.

### 3.1 A Unified Adversarial Game Framework for Evaluating Adversarial Robustness in Dataset Distillation

Previous studies on adversarial robustness in DD (Wu et al., 2024; Xue et al., 2024) have mainly focused on model accuracy, which provides an incomplete view of robustness. A more comprehensive evaluation should also consider attack effectiveness and time efficiency. To address this, we introduce a unified adversarial game framework that combines metrics for attack performance and efficiency, offering a more thorough assessment of how DD methods perform under various adversarial conditions.

**Definition 1** (Attacker Function). *Let $\mathcal{L} : \mathcal{Y} \times \mathcal{Y} \to \mathbb{R}$ be a loss function, and let $m \in \mathcal{M}$ be a model trained on a distilled dataset $\mathcal{S}$. The adversarial perturbation is constrained by $\|\hat{x} - x\|_p \leq \epsilon$. The attacker function $\mathcal{A} : \mathcal{X} \times \mathcal{Y} \times \mathcal{M} \to \mathcal{X}$ maps an input and a hypothesis to an adversarially perturbed version of the input, defined as:*

$$\mathcal{A}(x, y, m) = \underset{\|\hat{x}-x\|_p \leq \epsilon}{\arg\max} \, \mathcal{L}(m(x), y). \tag{1}$$

**Definition 2** (Defender Function). *Let $\mathcal{L} : \mathcal{Y} \times \mathcal{Y} \to \mathbb{R}$ be a loss function, and let $\mathcal{A}$ be a set of attacker functions with perturbation budget $\epsilon \in \mathcal{P}$. The defender function $\mathcal{D}$ aims to generate a synthetic dataset $\mathcal{S}$ such that the model $m$ trained on $\mathcal{S}$ minimizes the loss function against adversarial attacks from $\mathcal{A}$. Formally, $\mathcal{D}$ is defined as:*

$$\mathcal{D}(\mathcal{A}) = \arg\min_{\mathcal{S}} \max_{\mathcal{A}} \mathcal{L}(m(x), y), \tag{2}$$

*where $m \in \mathcal{M} = \mathcal{D}(\mathcal{T}, \mathcal{A})$ is the model trained on the dataset $\mathcal{S}$ generated by the defender function, and $\mathcal{A}$ represents the set of adversarial attacks.*

**Definition 3** (Attack Success Rate (ASR)). *Let $(x, y) \in (\mathcal{X}, \mathcal{Y})$ be an input-label pair, $a \in \mathcal{A}$ an adversarial attack function, and $m \in \mathcal{M}$ a model trained on a distilled dataset. The attack success rate (ASR) is defined as the probability that the model's prediction changes after an adversarial perturbation, specifically when the model correctly classifies the original input but misclassifies the perturbed one:*

$$\mathcal{ASR}(m; a) = \mathbb{E}_{(x,y) \in (\mathcal{X}, \mathcal{Y})} \mathbf{1}\{m(a(x)) \neq y \wedge m(x) = y\}, \tag{3}$$

*where $\mathbf{1}\{\cdot\}$ is the indicator function.*

**Definition 4** (Attack Success Time (AST)). *Let $(x, y) \in (\mathcal{X}, \mathcal{Y})$ be an input-label pair, and let $t$ denote the time taken to generate an adversarial example $\hat{x}$ using an attack function $a \in \mathcal{A}$ such that the model misclassifies the perturbed input, i.e., $m(\hat{x}) \neq y$. The Attack Success Time (AST) is defined as the expected time required for a successful adversarial attack:*

$$\mathcal{AST}(m; a) = \mathbb{E}_{(x,y) \in (\mathcal{X}, \mathcal{Y})} \left[ t \mid m(a(x)) \neq y \right]. \tag{4}$$

**Definition 5** (Adversarial Game Framework for Dataset Distillation against Multiple Attacks). *Given predefined thresholds $\gamma$ and $\beta$, which represent acceptable levels of attack success rate and time efficiency, respectively, and a set $\mathcal{A}$ of perturbation functions that may occur during test-time, the performance of the model is evaluated based on its attack success rate (ASR) and attack success time (AST) under these perturbations. The model is considered robust if:*

$$\frac{\mathbb{E}_{m \in \mathcal{M}} \mathbb{E}_{a \in \mathcal{A}} \mathcal{ASR}(m; a)}{\max\limits_{m^* \in \mathcal{M}, a^* \in \mathcal{A}} \mathcal{ASR}(m^*; a^*)} \leq \gamma \quad and \quad \frac{\mathbb{E}_{m \in \mathcal{M}} \mathbb{E}_{a \in \mathcal{A}} \mathcal{AST}(m; a)}{\max\limits_{m^* \in \mathcal{M}, a^* \in \mathcal{A}} \mathcal{AST}(m^*; a^*)} \geq \beta. \tag{5}$$

**Remark 1.** *In the adversarial game framework, the defender wins if two conditions are met: (1) the attack success rate $\mathcal{ASR}(m; a)$ is minimized below the threshold $\gamma$, and (2) the attack success time $\mathcal{AST}(m; a)$ is maximized above the threshold $\beta$. If either condition fails, the attacker wins. A win for the defender indicates effective robustness against adversarial perturbations, while a win for the attacker reveals vulnerabilities that need addressing. This game is defined over a set of models $m \in \mathcal{M}$, each trained on distinct distilled datasets $(\tilde{x}, \tilde{y}) \in (\tilde{\mathcal{X}}, \tilde{\mathcal{Y}}) \subseteq \mathcal{S}$, and subjected to various attacks $a \in \mathcal{A}$. When a specific model $m$ or attack $a$ is chosen, the multi-player game simplifies to a single-player game focused on their interaction.*

## 3.2 METRIC FOR EVALUATING ADVERSARIAL ROBUSTNESS

Building on the Definition 5 and Remark 1, we propose metrics that aggregate accuracy across both single and multiple attacks, as well as models trained on different IPC distilled datasets. This section introduces two key criteria for evaluating adversarial robustness: 1) attack effectiveness and 2) time efficiency, along with the corresponding metrics for each.

**Definition 6** (Robustness Ratio (RR)). *Given a neural network model $m \in \mathcal{M}$ and an adversarial attack function $a \in \mathcal{A}$, the robustness ratio is defined as:*

$$RR(m; a) = 100 \times \left[ 1 - \frac{\mathbb{E}_{m \in \mathcal{M}} \mathbb{E}_{a \in \mathcal{A}} \mathcal{ASR}(m; a)}{\max\limits_{m \in \mathcal{M}, a \in \mathcal{A}} \mathcal{ASR}(m; a)} \right]. \tag{6}$$

**Remark 2.** *The purpose of using "$1-$" in the formula is to emphasize model robustness rather than attack success. A higher attack success rate (ASR) indicates a more effective attack but a less robust model. Therefore, by subtracting the normalized attack success rate from 1, the formula inversely represents robustness. This way, when ASR is high, the robustness ratio (RR) will be low, and when ASR is low, the model is considered more robust. The formula also normalizes the ASR by dividing it by the maximum possible ASR to provide a standardized measure of robustness.*

**Definition 7** (Attack Efficiency Ratio (AE)). *Given a neural network model $m \in \mathcal{M}$ and an adversarial attack function $a \in \mathcal{A}$, the attack efficiency ratio is defined as:*

$$AE(m; a) = 100 \times \left[ \frac{\mathbb{E}_{m \in \mathcal{M}} \mathbb{E}_{a \in \mathcal{A}} \mathcal{AST}(m; a)}{\max\limits_{m \in \mathcal{M}, a \in \mathcal{A}} \mathcal{AST}(m; a)} \right]. \tag{7}$$

**Definition 8** (Comprehensive Robustness-Efficiency Index (CREI)). *The Comprehensive Robustness-Efficiency Index (CREI) integrates both the Robustness Ratio (RR) and the Attack Efficiency (AE) into a unified metric. It is defined as:*

$$CREI = \alpha \times RR + (1 - \alpha) \times AE, \tag{8}$$

*where $\alpha$ is an adjustable coefficient that determines the weighting between robustness and efficiency. This parameter allows for flexible balancing according to the specific needs of the evaluation.*

**Remark 3.** *The adversarial game framework can shift between multi-player and single-player scenarios, where "single-adversary" refers to a model facing one attack strategy, while "multi-adversary" involves multiple attack strategies. In this context, the metrics adjust: Robustness Ratio (RR) and Attack Efficiency (AE) become Single-Adversary Robustness Ratio (RRS) and Single-Adversary Attack Efficiency Ratio (AES) for single-adversary situations, and Multi-Adversary Robustness Ratio (RRM) and Multi-Adversary Attack Efficiency Ratio (AEM) for multi-adversary contexts. The defender aims to minimize the attack success rate, which aligns with maximizing RR, while optimizing AE corresponds to maximizing attack success time. Conversely, the attacker seeks to maximize AE and minimize RR. Analyzing these metrics within the framework allows for a clearer evaluation of dataset distillation robustness, highlighting the model's resilience against adversarial attacks and the efficiency of those attacks.*

## 4 ADVERSARIAL ROBUSTNESS BENCHMARK FOR DATASET DISTILLATION

### 4.1 OVERVIEW OF BEARD

BEARD consists of two main stages: the *Training Stage* and the *Evaluation Stage*, as illustrated in Figure 2. In the training stage (Section 4.1.1), models are trained on datasets from the dataset pool. The evaluation stage (Section 4.1.2) involves applying adversarial perturbations to test images from an attack library to assess model robustness. The benchmark comprises three key components: *Dataset Pool*, *Model Pool*, and *Evaluation Metrics*. More details are provided in Appendix A.

### 4.1.1 TRAINING STAGE

In the training stage, we focus on CIFAR-10 (Krizhevsky, 2009), CIFAR-100 (Krizhevsky, 2009), and TinyImageNet (Deng et al., 2009) due to their widespread use and diverse performance in dataset distillation (DD). Simpler datasets like MNIST (LeCun et al., 1998) and Fashion-MNIST (Xiao

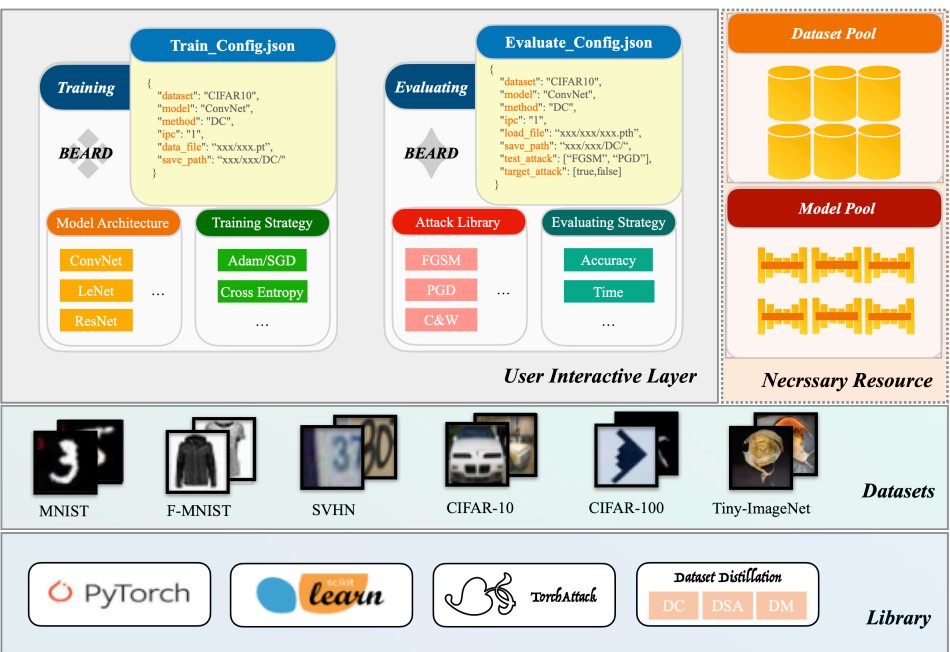

Figure 2: Illustration of BEARD: We first obtain a distilled dataset pool from the source dataset using various dataset distillation methods, such as DC (Zhao et al., 2020), DSA (Zhao & Bilen, 2023), DM (Zhao & Bilen, 2021), IDM (Zhao et al., 2023), BACON (Zhou et al., 2024a), among others. Next, we train neural networks on these diverse distilled datasets to generate a collection of pretrained models, forming our model pool. Finally, we evaluate the adversarial robustness of the models in the model pool by applying a variety of adversarial attack methods, including FGSM (Goodfellow et al., 2015b), PGD (Madry et al., 2018b), C&W (Carlini & Wagner, 2017), DeepFool (Moosavi-Dezfooli et al., 2016), AutoAttack (Croce & Hein, 2020), and others.

et al., 2017) are initially excluded but will be considered later to explore data efficiency. We evaluate six prominent DD methods: DC (Zhao et al., 2020), DSA (Zhao & Bilen, 2023), DM (Zhao & Bilen, 2021), MTT (Cazenavette et al., 2022), IDM (Zhao et al., 2023), and BACON (Zhou et al., 2024a), which represent a range of optimization techniques, including gradient matching (Zhao et al., 2020; Zhao & Bilen, 2023), distribution matching (Zhao & Bilen, 2021; Zhao et al., 2023), trajectory matching (Cazenavette et al., 2022), and optimization-based approaches (Zhou et al., 2024a). Synthetic datasets are generated using IPC-1, IPC-10, and IPC-50 settings, maintaining consistency with DC-bench (Cui et al., 2022) for hyperparameters. These datasets, produced by the six DD methods with diverse IPC settings, constitute the **Dataset Pool**, which is essential for evaluating the performance of various DD methods and ensuring a thorough comparison across different distillation approaches.

### 4.1.2 EVALUATING STAGE

In the evaluating stage, the **Model Pool** repository is utilized to streamline the assessment of robust models trained on distilled datasets. By integrating metrics derived from the adversarial game framework, including RR, AE, and CREI, this evaluation can more effectively measure the models' resilience against adversarial attacks within the competitive dynamics of the game setting. The repository facilitates the analysis of model performance and broader trends by consolidating checkpoints from various sources. However, challenges arise in unifying these models due to differing architectures and normalization techniques. After generating distilled datasets from the dataset pool, multiple models are trained from scratch using various distillation methods, IPC settings, and the Adam optimizer for 1,000 epochs. The models with the highest validation accuracy are selected and added to the model pool. Adversarial robustness is assessed using a diverse attack library compatible with Torchattacks (Kim, 2020), including methods like FGSM (Goodfellow et al., 2015b), PGD (Madry et al., 2018b), C&W (Carlini & Wagner, 2017), DeepFool (Moosavi-Dezfooli et al., 2016), and AutoAttack (Croce & Hein, 2020). Both targeted and untargeted attacks are conducted with a uniform perturbation budget of $|\epsilon| = \frac{8}{255}$ for most methods, with exceptions for DeepFool and C&W.

## 4.2 LEADERBOARDS

Figure 3: The top three entries on our CIFAR-10 leaderboard, with unified IPC settings, are available at `https://beard-leaderboard.github.io/`. The leaderboard utilizes metrics such as CREI, RRM, and AEM to assess robustness and attack efficiency. Additionally, it provides links to the code and distilled datasets for each entry, along with detailed information regarding authors, venues, and the last update.

We provide 12 leaderboards for CIFAR-10 (Krizhevsky, 2009), CIFAR-100 (Krizhevsky, 2009), and TinyImageNet (Deng et al., 2009), covering IPC-1, IPC-10, and IPC-50 settings. These leaderboards rank methods based on robustness and efficiency metrics, including RR, AE, and CREI. The leaderboard evaluates six dataset distillation methods: DC (Zhao et al., 2020), DSA (Zhao & Bilen, 2023), DM (Zhao & Bilen, 2021), MTT (Cazenavette et al., 2022), IDM (Zhao et al., 2023), and BACON (Zhou et al., 2024a). In terms of adversarial attacks, the leaderboards integrate various methods such as FGSM (Goodfellow et al., 2015b), PGD (Madry et al., 2018b), C&W (Carlini & Wagner, 2017), DeepFool (Moosavi-Dezfooli et al., 2016), and AutoAttack (Croce & Hein, 2020), all compatible with Torchattacks (Kim, 2020). Evaluating adversarial robustness is challenging due to the diversity of settings and attack types, and no unified evaluation framework currently exists. As illustrated in Figure 3, our leaderboards address this gap by providing a comprehensive evaluation of adversarial robustness in dataset distillation from a unified perspective.

## 5 ANALYSIS

### 5.1 ROBUSTNESS EVALUATION USING PROPOSED METRICS

The results in Figure 4 demonstrate that models trained on synthetic datasets generated by Dataset Distillation (DD) methods exhibit higher Multi-Adversary Robustness Ratio (RRM) under both targeted and untargeted adversarial attacks, although they show lower Multi-Adversary Attack Efficiency Ratio (AEM) compared to models trained on full-size datasets. Under targeted attacks, methods such as DSA, DM, and BACON demonstrate superior adversarial robustness, with RRM values increasing as dataset size expands, as shown in Figures 4 (a), (b), and (c). Conversely, untargeted attacks generally lead to a decline in RRM, but DSA, DM, BACON, and DC continue to perform robustly, as illustrated in Figures 4 (d), (e), and (f). The AEM metric further indicates that higher values correspond to increased time required by adversaries to attack the models, reflecting enhanced robustness. While models trained on full-size datasets tend to exhibit greater attack efficiency, the Comprehensive Robustness-Efficiency Index (CREI) highlights that DD methods, particularly DSA, DM, and BACON, achieve a more balanced performance in terms of both robustness and efficiency under targeted attacks. Despite smaller gains in untargeted attacks, DD methods consistently improve robustness across different datasets. More details are available in Appendix B.

### 5.2 ROBUSTNESS EVALUATION WITH DIVERSE IPCS

We find two key observations from Figure 5: 1) Increasing the IPC decreases adversarial robustness, as reflected by lower CREI values in Figures 5 (a), (b), and (c); and 2) Increasing the dataset scale enhances adversarial robustness when using DD methods compared to full-size datasets, as indicated by the distance between the black dashed line and the others in Figures 5 (d), (e), and (f). While full-size models often exhibit superior performance under targeted attacks, methods like BACON prove

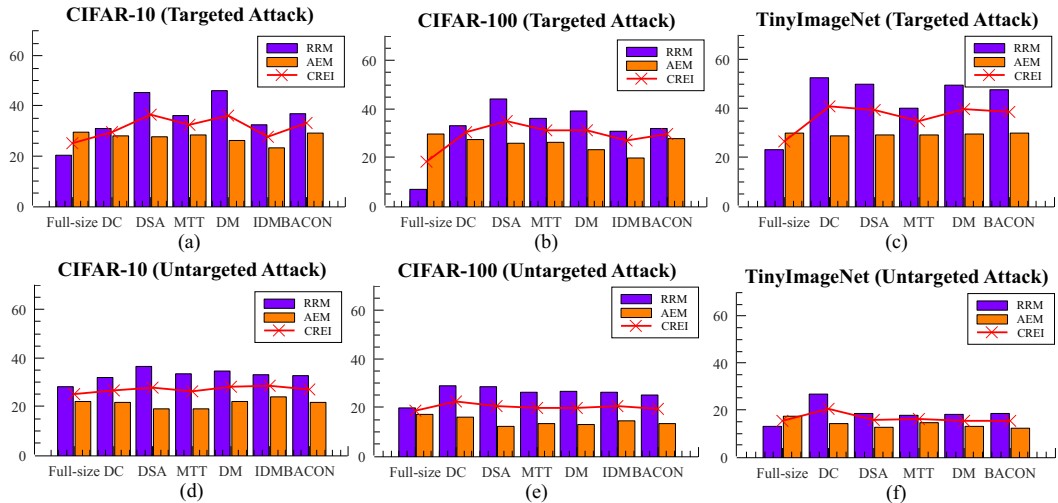

Figure 4: Performance of various dataset distillation methods under targeted and untargeted adversarial attacks on CIFAR-10, CIFAR-100, and TinyImageNet. The first row depicts targeted attacks with unified IPC settings, while the second row shows performance under untargeted attacks. Metrics used include Multi-Adversary Robustness Ratio (RRM), Multi-Adversary Attack Efficiency Ratio (AEM), and Comprehensive Robustness-Efficiency Index (CREI).

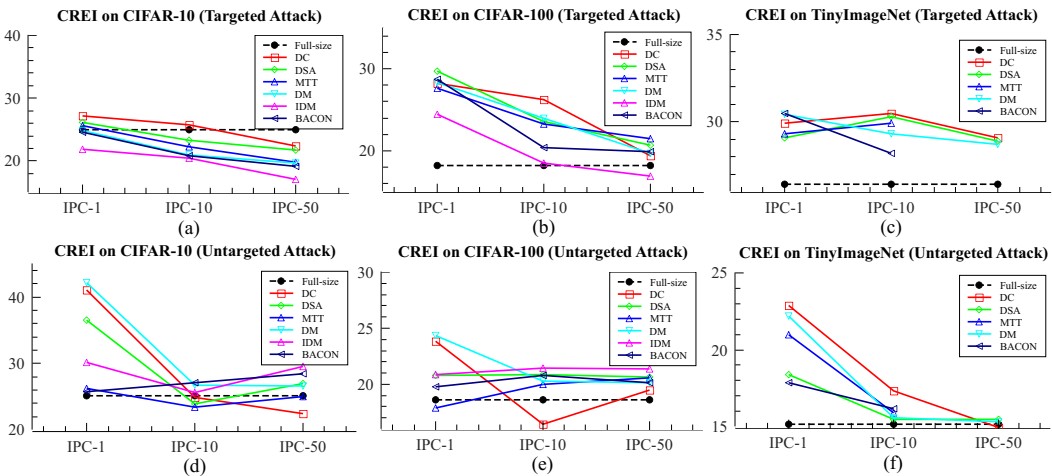

Figure 5: CREI trends under targeted and untargeted attacks across three datasets: CIFAR-10, CIFAR-100, and TinyImageNet. The x-axis represents the number of IPC, while the y-axis displays CREI values. Six DD methods (DC, DSA, MTT, DM, IDM, BACON) are compared to full-size datasets at IPC-1, IPC-10, and IPC-50, highlighting their robustness and efficiency across various attacks.

more effective with fewer images, highlighting their efficiency with smaller datasets. Additionally, the consistent CREI trends across CIFAR-10, CIFAR-100, and TinyImageNet indicate that these methods are robust and generalizable across different datasets. These findings underscore the complex relationship between IPC, dataset size, and adversarial robustness, affirming the effectiveness of specific methods in various scenarios. Further details are provided in Appendix B.

## 5.3 ROBUSTNESS EVALUATION WITH ADVERSARIAL TRAINING

Figure 6 demonstrates that Adversarial Training (AT) significantly enhances model robustness against both targeted and untargeted attacks within a unified IPC in a multi-adversary context. For targeted attacks, models utilizing AT (orange bars) achieve higher CREI values compared to those without AT (purple bars), particularly for methods such as DSA and BACON (Figure 6 (a)). In contrast, models

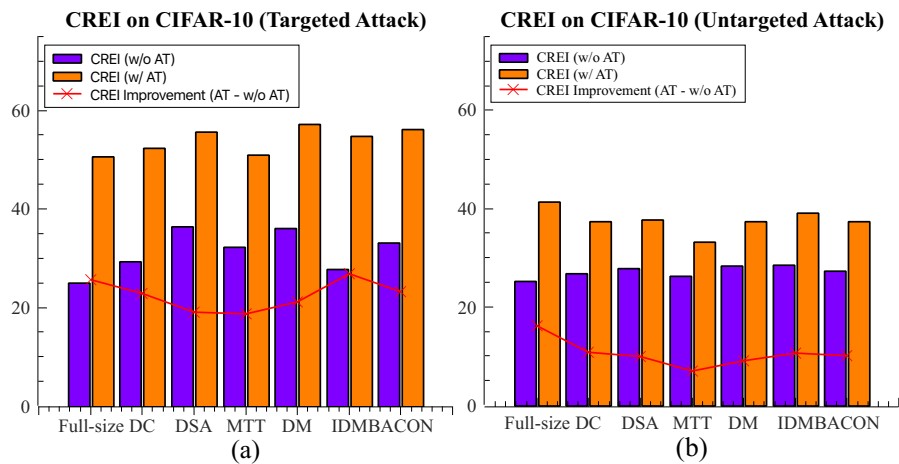

Figure 6: Illustration of CREI trends on CIFAR-10 under targeted and untargeted attacks with (w/) or without (w/o) Adversarial Training (AT). The x-axis shows DD methods (Full-size, DC, DSA, MTT, DM, IDM, BACON) under unified IPC, while the y-axis displays CREI values measuring adversarial robustness. CREI improvement indicates the difference between models with and without AT.

trained on full-size datasets exhibit lower CREI values, indicating reduced robustness, consistent with the trend of diminishing robustness as IPC increases. Without AT, all methods experience a significant decline in robustness, although DSA and DM perform relatively well. For untargeted attacks, the full-size dataset shows the most substantial improvement with AT, while all methods decline in performance without it (Figure 6 (b)). Notably, models trained on full-size datasets benefit more from AT than those trained on distilled datasets, suggesting that as dataset scale increases, the effectiveness of AT also increases, as illustrated by the red curve. These findings highlight the importance of selecting appropriate dataset scales to optimize the benefits of AT. Additional details can be found in Appendix B.

# 6 OUTLOOK

**Conclusion.** A standardized benchmark is crucial for advancing the evaluation of adversarial robustness in Dataset Distillation (DD) methods. We propose BEARD, an open and unified benchmark designed to assess adversarial robustness in DD. This benchmark includes a dataset pool, a model pool, and novel metrics (RR, AE, and CREI), and features a leaderboard that ranks models based on performance across three standard datasets under six adversarial attacks. Currently, the leaderboard includes 18 models trained on distilled datasets from six DD methods with three IPC settings. We aim to expand the benchmark by adding more distillation methods and larger datasets, as well as incorporating new and more potent attack types into our evaluation framework.

**Limitations and Future Plans.** While our benchmark currently encompasses six representative DD methods and six adversarial attack strategies, it covers most major types of both. Future plans include broadening BEARD to incorporate a wider array of DD methods, more sophisticated adversarial attack techniques, and larger datasets. Although the benchmark is focused primarily on enhancing the adversarial robustness of DD methods for image classification, we also intend to investigate effective strategies for attacking these methods and extend our evaluation to other modalities, such as text, graphs, and audio. This expansion could provide valuable insights into the adversarial robustness of distilled datasets across various domains.

**Practical Applications and Potential Impact.** The BEARD benchmark serves as both a research tool and a practical resource for evaluating the adversarial robustness of DD methods. By offering a standardized framework, BEARD helps users assess the strengths and weaknesses of various methods, thus supporting the development of more resilient data distillation techniques. Additionally, with increasing concerns about data security and privacy, BEARD holds considerable potential for applications in these critical areas, providing valuable insights into the robustness of distilled datasets in adversarial environments.

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

# Supplementary Material

### BEARD: Benchmarking the Adversarial Robustness in Dataset Distillation

- Appendix A offers an overview of BEARD, including the experimental setup, configurations, and implementation details.
- Appendix B provides a detailed analysis of robustness evaluation from three key perspectives: unified benchmarks, varying IPC settings, and the impact of adversarial training.

## A  OVERVIEW OF BEARD

### A.1  EXPERIMENTAL SETUP

#### A.1.1  DATASETS AND DISTILLATION METHODS

**Dataset.** In our experiments, we use three standard image classification datasets: CIFAR-10 (Krizhevsky, 2009), CIFAR-100 (Krizhevsky, 2009), and TinyImageNet (Deng et al., 2009). Each dataset has been selected for its relevance and complexity in the context of dataset distillation and adversarial robustness evaluation.

- **CIFAR-10** (Krizhevsky, 2009) contains 60,000 $32 \times 32$ color images in 10 classes, with 50,000 images for training and 10,000 for testing. The images are preprocessed to normalize pixel values to the range [0, 1].
- **CIFAR-100** (Krizhevsky, 2009) Similar to CIFAR-10 but with 100 classes, this dataset contains 60,000 images, divided into 50,000 training and 10,000 testing images. Each image is resized to $32 \times 32$ pixels and normalized.
- **TinyImageNet** (Deng et al., 2009) A subset of the large-scale ImageNet dataset, TinyImageNet contains 200 classes with 100,000 training images and 10,000 images each for validation and testing. Images are resized to $64 \times 64$ pixels and normalized.

**Dataset Distillation Methods.** Our benchmark evaluates six representative distillation methods: DC (Zhao et al., 2020), DSA (Zhao & Bilen, 2023), DM (Zhao & Bilen, 2021), MTT (Cazenavette et al., 2022), IDM (Zhao et al., 2023), and BACON (Zhou et al., 2024a). These methods represent a variety of optimization techniques commonly used in recent distillation research, including gradient matching (Zhao et al., 2020; Zhao & Bilen, 2023), distribution matching (Zhao & Bilen, 2021; Zhao et al., 2023), trajectory matching (Cazenavette et al., 2022), and Bayesian optimization-based approaches (Zhou et al., 2024a).

- **DC** (Zhao et al., 2020) formulates dataset distillation as a bi-level optimization problem, focusing on matching the gradients of deep neural networks trained on the original dataset $\mathcal{T}$ and the synthetic dataset $\mathcal{S}$.
- **DSA** (Zhao & Bilen, 2023) improves distillation by incorporating data augmentation, enabling the generation of more informative synthetic images, which enhances the performance of models trained with these augmentations.
- **DM** (Zhao & Bilen, 2021) offers a straightforward yet impactful method for generating condensed images by aligning the feature distributions of synthetic images $\mathcal{S}$ with those of the original training set $\mathcal{T}$ across multiple sampled embedding spaces.
- **MTT** (Cazenavette et al., 2022) introduces trajectory matching as a distillation technique, condensing large datasets into smaller ones by aligning the training trajectories of models trained on both the synthetic $\mathcal{S}$ and original $\mathcal{T}$ datasets.
- **IDM** (Zhao et al., 2023) proposes a novel dataset condensation approach based on distribution matching, which proves to be both efficient and promising for dataset distillation tasks.
- **BACON** (Zhou et al., 2024a) employs a Bayesian theoretical framework for dataset distillation, casting the problem as one of minimizing a risk function to significantly enhance distillation performance.

**Training Details.** Neural networks are trained from scratch on each distilled dataset, following a standardized training process across all experiments to ensure fair comparisons:

- **Optimizer:** The Adam optimizer is used with default settings, including a learning rate of 1e-4 and beta values of 0.9 and 0.999, ensuring stable and efficient optimization.
- **Epochs:** Models are trained for 1,000 epochs to ensure sufficient convergence and allow for the full learning potential of each distilled dataset.
- **Batch Size:** A batch size of 128 is employed to balance computational efficiency with model performance, optimizing resource usage without sacrificing accuracy.
- **Model Selection:** After training, the model with the highest validation accuracy on the original test set is selected and incorporated into the model pool for subsequent adversarial evaluations.

**Adversarial Attack Methods.** All attacks are implemented using the Torchattacks library (Kim, 2020), which includes a comprehensive set of current adversarial attack methods. To ensure fair comparisons, we apply consistent parameters across different models. Our attack library encompasses a range of methods, including FGSM (Goodfellow et al., 2015b), PGD (Madry et al., 2018b), C&W (Carlini & Wagner, 2017), DeepFool (Moosavi-Dezfooli et al., 2016), AutoAttack (Croce & Hein, 2020), and others. In the evaluation stage, adversarial perturbations are applied to assess the robustness of distilled datasets generated by various distillation methods. Both targeted and non-targeted attacks are performed to evaluate adversarial robustness. To ensure consistency, all trained models are subjected to identical parameters, with a perturbation budget set to $|\epsilon| = \frac{8}{255}$ for all methods except DeepFool and C&W.

- **FGSM** (Goodfellow et al., 2015b): Generates adversarial examples by perturbing the input in the direction of the gradient of the loss function, with a perturbation size set to $\epsilon = 8/255$.
- **PGD** (Madry et al., 2018b): Extends FGSM by applying iterative steps to create adversarial examples. The perturbation budget and step size are adjusted for each dataset to enhance attack strength.
- **C&W** (Carlini & Wagner, 2017): Focuses on optimizing adversarial examples to minimize perturbation while ensuring misclassification, providing a robust evaluation of model resilience.
- **DeepFool** (Moosavi-Dezfooli et al., 2016): Estimates the minimal perturbation required to induce misclassification, offering insights into the model's sensitivity to adversarial changes.
- **AutoAttack** (Croce & Hein, 2020): Combines multiple strong attacks to provide a comprehensive evaluation of model robustness, ensuring thorough assessment of adversarial resilience.

We generate synthetic images using 1, 10, and 50 images per class (IPC) from three datasets: CIFAR-10, CIFAR-100, and TinyImageNet. To assess the effectiveness of our approach, we train models on these synthetic images and evaluate their performance on the original test sets. All methods utilize the default data augmentation strategies provided by the original authors to ensure consistency in distillation performance evaluation. For a fair comparison in generalization, we use the synthetic datasets released by the authors.

After training, we apply a range of adversarial attacks to the models trained on the synthetic datasets and report the mean accuracy across 5 runs, with models randomly initialized and trained for 1,000 epochs. The evaluation metrics employed in our experiments are designed to provide a comprehensive assessment of adversarial robustness. These metrics include:

- **Single-Adversary Robustness Ratio (RRS):** Measures how effectively the models resist adversarial attacks under a single adversary.
- **Multi-Adversary Robustness Ratio (RRM):** Assesses the model's robustness against attacks from multiple adversaries.
- **Single-Adversary Attack Efficiency Ratio (AES):** Quantifies the efficiency of single adversarial attacks in terms of the time required to succeed.
- **Multi-Adversary Attack Efficiency Ratio (AEM):** Evaluates the efficiency of attacks involving multiple adversaries.

- **Comprehensive Robustness-Efficiency Index (CREI):** Integrates both robustness and attack efficiency into a unified metric, offering a balanced evaluation of model performance under adversarial conditions.

## A.2 EXPERIMENTAL SETTINGS

**Networks Architectures.** In our experiments, we employed the ConvNet architecture (Sagun et al., 2017) for dataset distillation, following methodologies from prior studies, including DC-bench (Cui et al., 2022) and BACON (Zhou et al., 2024a). The ConvNet consists of three identical convolutional blocks followed by a final linear classifier. Each block features a convolutional layer with 128 kernels of size $3 \times 3$, instance normalization, ReLU activation, and average pooling with a stride of 2 and a pooling size of $3 \times 3$. This architecture configuration is consistent with the settings outlined in DC-bench and BACON, ensuring adherence to established practices in dataset distillation.

**Evaluation Protocol.** We generate synthetic images using 1, 10, and 50 images per class (IPC) from three datasets: CIFAR-10, CIFAR-100, and TinyImageNet. To assess the effectiveness of our approach, we train models on these synthetic images and evaluate their performance on the original test sets. All methods utilize the default data augmentation strategies provided by the original authors to maintain consistency in distillation performance evaluation. For fair comparisons in generalization, we employ the synthetic datasets released by the authors.

Following model training, we apply adversarial attacks to evaluate the robustness of the models trained on the various synthetic datasets. We report the mean accuracy across 5 runs, with models randomly initialized and trained for 1,000 epochs. The performance is measured using the condensed set as the primary evaluation metric.

## A.3 IMPLEMENTATION DETAILS

The BEARD benchmark builds upon the software foundation established by BACON (Zhou et al., 2024a). For generating synthetic images in the dataset pool, we use the Stochastic Gradient Descent (SGD) optimizer with a learning rate of 0.2 and a momentum of 0.5, applied to synthetic datasets containing 1, 10, and 50 images per class (IPC). In the subsequent model training phase, we employ the same SGD optimizer, but adjust the learning rate to 0.01, momentum to 0.9, and apply a weight decay of 0.0005. The batch size is set to 256. All experiments, including both the generation of synthetic datasets and the training of models, are conducted using NVIDIA RTX 2080 Ti GPU clusters. Additionally, we provide a configuration JSON file to facilitate the convenient setup and management of experimental parameters.

## B ANALYSIS

### B.1 ROBUSTNESS EVALUATION USING RR, AE, AND CREI METRICS

The Table 1 compares the performance of various dataset distillation methods using three key metrics: Multi-Adversary Robustness Ratio (RRM), Multi-Adversary Attack Efficiency Ratio (AEM), and Comprehensive Robustness-Efficiency Index (CREI). The evaluation covers both targeted and untargeted adversarial attacks across three datasets: CIFAR-10, CIFAR-100, and TinyImageNet.

**Targeted Attacks.** Dataset distillation methods demonstrate substantial improvements in robustness compared to full-size models. For example, in CIFAR-10, DM achieves a RRM of 46.01%, a significant increase from the 20.42% of the full-size model. Similarly, DSA achieves a RRM of 45.22%. These enhancements are evident across CIFAR-100 and TinyImageNet, where DM and DSA continue to outperform full-size models. For instance, in CIFAR-100, DM has a RRM of 39.32%, compared to 6.77% for the full-size model. On TinyImageNet, DM achieves a RRM of 49.57%, compared to 22.99% for the full-size model. Despite these improvements in robustness, distillation methods like DC and DSA show a slight reduction in AEM values. For example, in CIFAR-10, DC has an AEM of 27.91% and DSA has 27.64%, compared to 29.39% for the full-size model. This indicates a trade-off between robustness and efficiency. The CREI scores further illustrate this balance: DM and DSA achieve high CREI values, with DM reaching 36.01% in CIFAR-10 and DSA 36.43%, showcasing their effective trade-off between robustness and efficiency.

Table 1: Performance comparison of dataset distillation methods under various adversarial attacks. Metrics include Multi-Adversary Robustness Ratio (RRM), Multi-Adversary Attack Efficiency Ratio (AEM), and Comprehensive Robustness-Efficiency Index (CREI). The Targ. Att. and Untarg. Att. denote the Targeted Attack and Untargeted Attack, respectively.

| Evaluation | | | Dataset Distillation (%) | | | | | | |
|---|---|---|---|---|---|---|---|---|---|
| Metric | Attack Type | Dataset | Full-size | DC | DSA | MTT | DM | IDM | BACON |
| RRM | Targ. Att. | CIFAR-10 | 20.42 | 30.79 | 45.22 | 36.00 | 46.01 | 32.35 | 36.83 |
| | | CIFAR-100 | 6.77 | 33.11 | 43.97 | 36.06 | 39.32 | 30.79 | 31.81 |
| | | TinyImageNet | 22.99 | 52.62 | 49.87 | 40.05 | 49.57 | / | 47.57 |
| | Untarg. Att. | CIFAR-10 | 20.42 | 30.79 | 45.22 | 36.00 | 46.01 | 32.35 | 36.83 |
| | | CIFAR-100 | 6.77 | 33.11 | 43.97 | 36.06 | 39.32 | 30.79 | 31.81 |
| | | TinyImageNet | 22.99 | 52.62 | 49.87 | 40.05 | 49.57 | / | 47.57 |
| AEM | Targ. Att. | CIFAR-10 | 29.39 | 27.91 | 27.64 | 28.52 | 26.01 | 23.15 | 29.27 |
| | | CIFAR-100 | 29.59 | 27.50 | 26.05 | 26.25 | 23.31 | 19.89 | 27.76 |
| | | TinyImageNet | 29.83 | 28.80 | 28.97 | 29.26 | 29.55 | / | 29.96 |
| | Untarg. Att. | CIFAR-10 | 21.91 | 21.53 | 18.97 | 19.21 | 22.13 | 23.89 | 21.53 |
| | | CIFAR-100 | 17.29 | 16.06 | 12.26 | 13.23 | 12.83 | 14.44 | 13.34 |
| | | TinyImageNet | 17.31 | 14.04 | 12.77 | 14.71 | 13.08 | / | 12.09 |
| CREI | Targ. Att. | CIFAR-10 | 24.91 | 29.35 | 36.43 | 32.26 | 36.01 | 27.75 | 33.05 |
| | | CIFAR-100 | 18.18 | 30.31 | 35.01 | 31.16 | 31.32 | 27.16 | 29.78 |
| | | TinyImageNet | 26.41 | 40.71 | 39.42 | 34.66 | 39.56 | / | 38.76 |
| | Untarg. Att. | CIFAR-10 | 25.12 | 26.70 | 27.75 | 26.26 | 28.32 | 28.46 | 27.20 |
| | | CIFAR-100 | 18.60 | 22.40 | 20.40 | 19.65 | 19.78 | 20.36 | 19.30 |
| | | TinyImageNet | 15.15 | 20.46 | 15.67 | 16.13 | 15.51 | / | 15.24 |

**Untargeted Attacks.** The robustness improvements with distillation methods are less pronounced compared to targeted attacks. For instance, in CIFAR-10, while DM and DSA still offer high RRM (45.22% and 46.01%, respectively), the gap between these methods and full-size models is narrower. The AEM values for distillation methods are generally lower, indicating that these methods require less time and computational resources for adversarial attacks compared to full-size models. For example, the AEM for DC in CIFAR-10 under untargeted attacks is 21.53%, compared to 21.91% for the full-size model. Similarly, DSA shows an AEM of 18.97% in CIFAR-10, which is lower than the 21.91% for the full-size model. The CREI scores reflect this trend, with methods like DM and DSA achieving reasonable CREI values, such as 27.75% for DM in CIFAR-10, demonstrating a balanced performance between robustness and efficiency despite the slight trade-off in robustness.

B.2 ROBUSTNESS EVALUATION WITH DIVERSE IPCS

The robustness evaluation, conducted across targeted and untargeted attacks, reveals two key observations: 1) increasing the number of images per class (IPC) decreases adversarial robustness, as evidenced by lower CREI values across various methods and datasets; and 2) increasing the dataset scale enhances adversarial robustness when using dataset distillation methods, particularly when comparing distilled datasets to full-size datasets.

**Targeted Attacks.** In the context of targeted attacks, increasing IPC values typically leads to reduced adversarial robustness, as seen from the declining CREI scores. For instance, on CIFAR-10, methods like DC and DSA perform best at IPC = 1, showing strong robustness, but their performance decreases with larger IPC values. Similarly, for CIFAR-100, BACON outperforms other methods at IPC = 1, though its robustness diminishes at higher IPC levels. Importantly, as the dataset scale increases, the advantage of dataset distillation methods over Full-size datasets becomes more pronounced. For example, in TinyImageNet at IPC = 1, DC and DSA maintain high CREI scores, surpassing the Full-size model, emphasizing that dataset distillation methods can achieve better robustness with smaller dataset sizes under targeted attacks. The detailed results are presented in Table 2.

**Untargeted Attacks.** Under untargeted attacks, the trend of decreasing robustness with increasing IPC is also observed, but the effects are less severe compared to targeted attacks. For CIFAR-10, DC

Table 2: Comparison of adversarial robustness using CREI for different dataset distillation methods under targeted attacks across various datasets and IPC settings.

| Dataset | IPC | Dataset Distillation (%) | | | | | |
|---|---|---|---|---|---|---|---|
| | | DC | DSA | MTT | DM | IDM | BACON |
| CIFAR-10 | Full-size | | | 24.91 | | | |
| | 1 | 27.19 | 26.11 | 25.67 | 24.87 | 21.92 | 24.62 |
| | 10 | 25.73 | 23.31 | 22.21 | 20.90 | 20.41 | 20.83 |
| | 50 | 22.36 | 21.75 | 19.82 | 19.70 | 17.11 | 19.10 |
| CIFAR-100 | Full-size | | | 18.18 | | | |
| | 1 | 28.23 | 29.74 | 27.65 | 28.30 | 24.44 | 28.71 |
| | 10 | 26.23 | 23.58 | 23.23 | 23.97 | 18.47 | 20.38 |
| | 50 | 19.40 | 20.64 | 21.51 | 19.70 | 16.86 | 19.86 |
| TinyImageNet | Full-size | | | 26.41 | | | |
| | 1 | 29.94 | 29.08 | 29.33 | 30.44 | / | 30.50 |
| | 10 | 30.46 | 30.28 | 29.93 | 29.30 | / | 28.18 |
| | 50 | 29.10 | 28.89 | / | 28.72 | / | / |

and DM perform strongly at IPC = 1, with DC achieving the highest CREI score. Notably, as the dataset size increases (e.g., TinyImageNet), the gap in robustness between distillation methods and Full-size datasets becomes more evident. For instance, DC consistently outperforms the Full-size model in TinyImageNet at IPC = 1, while showing comparable or better robustness even at higher IPC values. This reinforces the observation that dataset distillation methods not only excel with fewer images per class but also offer greater robustness in larger datasets, especially when facing untargeted attacks. Detailed results are provided in Table 3.

Table 3: Comparison of adversarial robustness using CREI for different dataset distillation methods under untargeted attacks across various datasets and IPC settings.

| Dataset | IPC | Dataset Distillation (%) | | | | | |
|---|---|---|---|---|---|---|---|
| | | DC | DSA | MTT | DM | IDM | BACON |
| CIFAR-10 | Full-size | | | 25.12 | | | |
| | 1 | 41.11 | 36.59 | 26.25 | 42.21 | 30.18 | 25.79 |
| | 10 | 24.85 | 23.90 | 23.40 | 26.73 | 25.60 | 27.09 |
| | 50 | 22.50 | 27.00 | 25.06 | 26.61 | 29.59 | 28.48 |
| CIFAR-100 | Full-size | | | 18.60 | | | |
| | 1 | 23.87 | 20.81 | 17.91 | 24.32 | 20.85 | 19.76 |
| | 10 | 16.44 | 20.83 | 19.97 | 20.28 | 21.43 | 20.78 |
| | 50 | 19.46 | 20.67 | 20.54 | 20.10 | 21.34 | 20.11 |
| TinyImageNet | Full-size | | | 15.15 | | | |
| | 1 | 22.86 | 18.41 | 20.98 | 22.20 | / | 17.88 |
| | 10 | 17.33 | 15.50 | 15.90 | 15.59 | / | 16.18 |
| | 50 | 14.96 | 15.50 | / | 15.30 | / | / |

## B.3    ROBUSTNESS EVALUATION WITH ADVERSARIAL TRAINING

**Targeted Attacks.**    Table 4 shows that adversarial training notably enhances robustness under targeted attacks. Methods like BACON and DM achieve high CREI values of 56.18% and 57.21%, respectively, indicating superior robustness compared to other methods. The Full-size dataset, despite being complete, exhibits lower robustness with a CREI of 50.54%, underscoring the effectiveness of distillation techniques in improving adversarial resilience.

**Untargeted Attacks.**    In the context of untargeted attacks, the Full-size dataset achieves a CREI of 41.33%, slightly outperforming other methods. BACON and DM demonstrate similar performance with CREI values of 37.25% and 37.34%, respectively. Without adversarial training, all methods

Table 4: Comparison of adversarial robustness using CREI for different dataset distillation methods with and without adversarial training under targeted and untargeted attacks.

| Attack Type | Adversarial Training | Dataset Distillation (%) | | | | | | |
|---|---|---|---|---|---|---|---|---|
| | | Full-size | DC | DSA | MTT | DM | IDM | BACON |
| Targeted | w/ AT | 50.54 | 52.30 | 55.56 | 50.96 | 57.21 | 54.67 | 56.18 |
| | w/o AT | 24.91 | 29.35 | 36.43 | 32.26 | 36.01 | 27.75 | 33.05 |
| Untargeted | w/ AT | 41.33 | 37.39 | 37.59 | 33.13 | 37.34 | 39.05 | 37.25 |
| | w/o AT | 25.12 | 26.70 | 27.75 | 26.26 | 28.32 | 28.46 | 27.20 |

show reduced robustness, with DSA and DM maintaining relatively higher CREI values of 27.75% and 28.32%. These findings highlight the crucial role of adversarial training in enhancing robustness and confirm the advantages of dataset distillation methods in enhancing adversarial robustness.

