# OpenReview forum: "BEARD: Benchmarking the Adversarial Robustness for Dataset Distillation"
_ICLR.cc/2025/Conference — ICLR 2025 Conference Withdrawn Submission_

### Official Review · Reviewer_fu4Q · 2024-11-03

**Soundness:** 4
**Presentation:** 4
**Contribution:** 2
**Rating:** 5
**Confidence:** 3

**Summary:**

BEARD proposes a standardized benchmarking tool for the evaluation of adversarial robustness of models trained with dataset distillation (DD) techniques. BEARD implements several existing dataset distillation methods as baselines, as well as classic adversarial attack strategies for evaluation. BEARD additionally introduces three metrics to compare performance of different DD methods.

**Strengths:**

- Provides a standardized/common notation under which to describe adversarial attack tactics and dataset distillation strategies
- Multiple useful evaluation metrics RR, AE, CREI for comparing different strategies
- Implementation of a comprehensive framework combining libraries for network training, adversarial attack evaluation, and dataset distillation
- Novel observations like increased IPC leading to decreasing adversarial robustness, but increasing dataset scale enhancing adversarial robustness which may not be an immediately obvious conclusion

**Weaknesses:**

- The implemented library serves mostly as a glue library for established packages (TorchAttack and dataset distillation package)
- The proposed metrics are straightforward modifications of existing metrics (thresholding ASR and AST)

**Questions:**

- The code provided uses pre-distilled datasets, what procedure/libraries/packages were used for this distillation?
- What is your procedure to choose an appropriate &gamma; and &Beta; in the adversarial game framework?

---

### Official Review · Reviewer_FPZb · 2024-11-03

**Soundness:** 2
**Presentation:** 2
**Contribution:** 2
**Rating:** 5
**Confidence:** 4

**Summary:**

This paper proposes a dataset distillation benchmark called BREAD, designed to evaluate the robustness of various existing dataset distillation methods. The authors introduce several metrics to support this benchmark. For instance, the Attack Success Time (AST) metric measures the time taken to generate a successful adversarial example, while the Robustness Ratio (RR) assesses robustness across different model architectures and attack methods.

However, the paper lacks novelty, and its contribution is limited. The metrics provided are not specifically tailored for dataset distillation tasks. More importantly, an existing study by Wu et al. (2024) has already proposed a robustness benchmark, predating this work. I do not see any significant improvement in this paper; the authors merely note that the previous work does not release its code. However, this paper also does not append their codes as supplementary.

**Strengths:**

Strength:
1. The authors append the link of their github project page to demonstrate their leadboard. The dataset pool and model pool contains the codes of the dataset distillation methods and model architectures.

2. The authors provide many figures to demonstrate their proposed metrics.

**Weaknesses:**

Weakness:
1. The github project page is not completed. The codes is not released and only 6 methods are listed in the leaderboards.

2. Although the paper is 19 pages long, the benchmark neglects several aspects:
(a) The robustness evaluation considers only white-box attacks, omitting black-box attacks.
(b) Only three datasets are considered, CIFAR10, CIFAR100, and Tiny-Imagenet. The imagenet subsets and ImageNet-1k are also very important in dataset distillation tasks.
(c) Many methods are not evaluated in this benchmark, including DATM, DREAM.

[1]Ziyao Guo, Kai Wang, George Cazenavette, HUI LI, Kaipeng Zhang, and Yang You. Towards
lossless dataset distillation via difficulty-aligned trajectory matching. In International Conference on Learning Representations (ICLR), 2024.

[2] Liu, Y., Gu, J., Wang, K., Zhu, Z., Jiang, W., & You, Y. (2023). Dream: Efficient dataset distillation by representative matching. In Proceedings of the IEEE/CVF International Conference on Computer Vision (pp. 17314-17324).

3. The motivation of this paper is unclear. What specific contributions set this work apart from the existing study by Wu et al. (2024)? The authors merely criticize that the previous work does not release its code; however, they also do not release their own code.

4. Attack Success Time (AST) defined here is not inappropriate. The inference time varies among different models and different attack methods. The metric should only vary in different distilled datasets.

5. What is the purpose to demonstrating the figure 2? There is no useful information disclosed in figure 2, it is better to place it in the appendix.

**Questions:**

As stated in weakness part. To conclude, this work resembles engineering more than research. I hope the authors could help me address my concerns in weakness part.

---

### Official Review · Reviewer_B4VD · 2024-11-05

**Soundness:** 2
**Presentation:** 2
**Contribution:** 2
**Rating:** 5
**Confidence:** 4

**Summary:**

This paper introduces BEARD, a comprehensive benchmark framework for evaluating adversarial robustness of Dataset Distillation (DD) methods, featuring novel evaluation metrics and an adversarial game framework. The authors provide extensive experiments across multiple datasets and DD methods, along with a public model pool, dataset pool and leaderboard. The work establishes a standardized evaluation protocol for DD methods' robustness, though several aspects require further investigation.

**Strengths:**

1. Addresses an important gap in DD research by providing a standardized way to evaluate adversarial robustness
2. Comprehensive experimental evaluation across multiple datasets, methods and attack types
3. Open-source implementation and leaderboard to benefit the research community

**Weaknesses:**

- Only considers image classification tasks, limiting generalizability
- Uses relatively small datasets (CIFAR-10/100, TinyImageNet) compared to modern standards
- Limited model architectures (only ConvNet) without exploring more modern architectures like Vision Transformers
- No analysis of trade-offs between robustness and model performance

**Questions:**

- How does computational complexity scale with dataset size and model architecture? Is BEARD practical for evaluating large-scale models and datasets?
- For a comprehensive benchmark, have you considered combining DD methods with defense methods? The interaction between DD and defenses could provide valuable insights.

---

### Official Review · Reviewer_393r · 2024-11-06

**Soundness:** 3
**Presentation:** 3
**Contribution:** 2
**Rating:** 5
**Confidence:** 4

**Summary:**

This paper introduced a benchmark that employs an adversarial game framework to systematically evaluate dataset distillation (DD) models under various adversarial attack scenarios. The authors explored the vulnerabilities introduced by adversarial attacks, and the robustness of models trained on distilled datasets.

**Strengths:**

- This paper proposed three novel metrics - Robustness Ratio (RR), Attack Efficiency Ratio (AE), and Comprehensive Robustness-Efficiency Index (CREI) - to evaluate adversarial robustness against different attacks.
- It established a leaderboard that ranks existing DD methods based on these metrics;
- It provided open-source code with comprehensive documentation, along with a Model Pool and Dataset Pool to facilitate adversarial robustness evaluations.

**Weaknesses:**

This paper (BEARD) seems to overlap significantly with DD-RobustBench [Wu et al., 2024], including:
- The goal of both papers is the same: benchmark adversarial robustness in dataset distillation
- They both use a similar three-stage evaluation pipeline: distillation -> model training -> adversarial attack testing
- They both adopt a similar experimental setup, including datasets, IPC settings, attach methods, DD methods (DD-RobustBench tested more recent DD methods, e.g. D4M, which is particularly important as it represents the latest generative dataset distillation method that achieves high scalability and accuracy).
- The analysis perspectives are very similar, for example, they both analyzed performance across different IPCs.

**Questions:**

It would be great to emphasize the unique novelty to differentiate from existing literature given the significant overlap with DD-RobustBench [Wu et al., 2024]. Especially, there are misunderstandings the authors may have about DD-RobustBench:
- The authors of DD-RobustBench did provide their code on GitHub.
- Beyond the model accuracy, DD-RobustBench also introduced the Drop Rate (DR) metric specifically designed to evaluate robustness and conducts a comprehensive analysis on how different components affect robustness.

---

### Note · Authors · 2024-11-14

**Comment:**

We sincerely thank the reviewer for their invaluable feedback and time. We will incorporate the provided comments for our next submission.

**Withdrawal Confirmation:**

I have read and agree with the venue's withdrawal policy on behalf of myself and my co-authors.